# Effect of relative weight limit set as a body weight percentage on work-related low back pain among workers

**Kazuyuki Iwakiri**[ORCID]*, **Takeshi Sasaki, Midori Sotoyama, Tanghuizi Du, Keiichi Miki, Fuyuki Oyama**

National Institute of Occupational Safety and Health, Kawasaki, Japan

* iwakiri@h.jniosh.johas.go.jp

## Abstract

### Introduction

A quarter of work-related low back pain (LBP) cases result from handling heavy loads in Japan. The maximum weight male/female workers can handle is 40%/24% of their body weight but has set a constant load weight in ISO 11228–1 and NIOSH lifting equation. The preventive effect of the relative weight limit on LBP has not been clarified. This study aimed to identify the effect of relative weight limits set as body weight percentages on LBP prevalence.

### Methods

Data from 21924 workers were collected via a web-based survey in 2022. The workers were categorized into three groups: group A, "no handling," group B, "handling loads up to 40%/24% or less of body weight," and group C, "handling loads over 40%/24% of body weight." Moreover, they were categorized into eight groups: no handling, 1–5 kg, 5–10 kg, 10–15 kg, 15–20 kg, 20–25 kg, 25–30 kg, and ≥30 kg. Multiple logistic regression analysis was used to identify the effects of the limits set to body weight percentages and constant load weights on LBP.

### Results

In groups A, B, and C, 25.5%, 39.2%, and 47.3% of males or 16.9%, 26.4%, and 38.0% of females had LBP, respectively. The odds ratio (OR) of LBP was significantly greater in group B than in group A and even greater in group C. The OR of LBP among workers handling loads under 10 kg was not significantly different compared to no-handling workers.

### Conclusions

LBP prevalence was greater in group B than in group A but lesser than in group C. Weight limits based on body weight percentages could not eliminate the factor of handling loads. However, handling loads under 10 kg suppressed LBP. Relative weight limits set as body weight percentages were inappropriate and ineffective for preventing LBP.

**Data Availability Statement:** All relevant data are within the paper and its Supporting information files.

**Funding:** This study was supported by the National Institute of Occupational Safety and Health, Japan (N-P03-01).The funders had no role in study design, data collection and analysis, decision to publish, or preparation of the manuscript.

**Competing interests:** The authors have declared that no competing interests exist.

## Introduction

Although conflicting evidence has been found regarding the association between work-related low back pain (LBP) and manual handling [1–3], manual handling of heavy loads and materials is a possible risk factor for LBP [4–10]. In Japan, approximately 5000 cases of work-related LBP are recognized as industrial accidents every year, and a quarter of these cases are caused by handling heavy loads [11]. Therefore, as a preventive measure, in the Guidelines on the Prevention of Lumbago in the Workplace (lumbago guidelines), male and female workers can handle loads up to 40% and 24% of their body weight (60% of males), respectively [12]. Moreover, in the Regulations on Labor Standards for Women, female workers can handle loads up to 20 kg in continuous work and 30 kg in intermittent work [13]. The upper limit of load weight (weight limit) for female workers is the lower value of either guidelines or regulations, while the weight limit for male workers follows the lumbago guidelines.

The suggested weight limit of manual handling in Japan is inconsistent with international guidelines. According to the lumbago guidelines [12], a Japanese adult male with an average body weight of 67.4 kg [14] can handle loads up to 27.0 kg. However, International Organization for Standardization (ISO) 11228–1 [15] and Health and Safety Executive (HSE) guide [16] have established a weight limit of 25 kg. National Institute for Occupational Safety and Health (NIOSH) lifting equation [17] has established a weight limit of 23 kg, regardless of body weight. In addition, these have calculated a handling load considering work contents based on the weight limits. Most Japanese males are smaller than Western males but will be allowed to handle heavier load weights. A Japanese adult female with an average body weight of 53.6 kg [14] is suggested to handle loads up to 12.9 kg. This weight is lighter than the weight limits of 20 kg in ISO 11228–1 [15] and 16 kg in the HSE guide [16]. However, the incidence of LBP among Japanese females has increased recently [11].

Moreover, biomechanical studies have shown that limiting load weights as body weight percentages are problematic. The compression and shearing forces on the lumbar disc are greater in a heavier person than in a lighter person in the case of two people of the same height [18, 19]. According to the weight limit set as body weight percentages, a heavier person can lift heavier loads, increasing the shear force on the lumbar disc.

Therefore, the relative weight limit based on body weight percentages may not suppress LBP prevalence. However, no studies have examined the effectiveness of relative weight limits on LBP. This study aimed to identify the effect of relative weight limits set as body weight percentages on LBP prevalence.

## Methods

### Research design

This cross-sectional survey was conducted over the internet. Participants were Japanese male and female workers aged 20 to 75 years who were working or taking a leave of absence due to work-related LBP in four industries: "manufacturing," "wholesale and retail trade," "construction," and "transport and postal activities." These four industries were selected as workers in these domains handle heavy loads and materials regularly. The total working population in the four industries was 29.4 million [20]. Among them, data were collected from a total of 30000 workers, 7500 workers per industry, according to the sex and age distribution of the Labor Force Survey [20].

## Questionnaire

The questionnaire gathered information on basic demographic and job characteristics, job stressors, LBP severity, working posture, manual handling status, and load weight handled regularly. The time point of questions from LBP severity to load weight was defined as when a worker had the first incidence of LBP at the current job. Moreover, we questioned the participants regarding the impact of COVID-19 on their working style and work content.

**Basic information.** Basic demographic and job characteristics included sex, age, body height and weight, body mass index, smoking status, industry, work time and shifts, and the total number of working hours per week.

**Job stressor.** Questions on job stressors were developed based on the three job demand items, three job control items, and six worksite social support items of the Brief Job Stress Questionnaire [21, 22]. Job demand parameters were assessed by "I have an extremely large amount of work to do," "I can't complete work in the required time," and "I have to work as hard as I can." Job control parameters were assessed by "I can work at my own pace," "I can choose how and in what order to do my work," and "I can reflect my opinion on workplace policy." Worksite social support parameters were assessed by "How freely can you talk with the following people?" "How reliable are the following people when you are troubled?" and "How well will the following people listen to you when you ask for advice on personal matters?" The people were the respondent's superiors and coworkers. Responses were measured using the four-point scale. Total scores for job demand and control responses ranged from 3 to 12; 3–7 points were defined as "low stress," and 8–12 points were defined as "high stress." For the six worksite social support responses, scores ranged from 6 to 24; 6–15 points were defined as "low stress," and 16–24 points were defined as "high stress."

**Low back pain.** LBP was defined as pain in the lower back or buttocks lasting more than a day, including pain and numbness in the legs, which was the first incidence at the current job; however, this did not include back pain during menstruation, pregnancy or a cold. LBP severity was divided into four grades based on the scheme devised by Von Korff et al. [23]: grade 0, no LBP; grade 1, LBP not interfering with work; grade 2, LBP interfering with work; grade 3, LBP interfering with work and leading to sick leave. Grades 0 and 1 were defined as "non-severe LBP," whereas grades 2 and 3 were defined as "severe LBP." The prevalence of severe LBP was calculated as a percentage of LBP caused by career-time cumulative burden. In addition, we questioned how long ago LBP had occurred.

**Working posture and manual handling status.** Questions regarding working posture included working posture and postural change of the body trunk. These parameters were assessed as follows: working posture—neutral posture, forward-bending position, a half-crouching position, twisting posture, unstable posture, and other postures; postural change—complete freedom, mostly free, a little freedom, and no freedom. The neutral posture was defined as a straight-backed posture without excessive force, whereas the non-neutral posture was described as other unnatural positions.

Questions about manual handling status included handling situation, handling time and the number of times per day, and average carrying distance: handling situation—no handling, lifting and lowering, carrying, pushing, pulling, rolling, and others; handling time—<1 hour, 1–2 hours, 2–3 hours, 3–4 hours, and ≥4 hours; the number of times per day to handle a load by hand—<3 times, 3–5 times, 5–10 times, 10–30 times, and ≥30 times; average carrying distance—no carrying, 1–5 m, 5–10 m, 10–20 m, and ≥20 m. For working posture and handling situations, multiple answers were allowed.

**Weight of load.** Load weight was defined as the weight value per person handled by human power regularly. Load weight was classified into three categories: no handling (group

A), handling loads up to 40% for male workers or 24% for female workers or less of body weight (group B), and handling loads over 40% for male workers or 24% for female workers of body weight (group C). Moreover, load weight was classified into eight categories: no handling, 1–5 kg, 5–10 kg, 10–15 kg, 15–20 kg, 20–25 kg, 25–30 kg, and ≥30 kg.

## Survey procedure

The web-based questionnaire was answered by workers registered with multiple monitor research companies via an internet research company. Data collection started in early January 2022, ending sequentially when the number of participants in each industry reached 7500 and was completed in late January 2022.

The workers were fully informed of the study plan and assured that the personal information provided would be kept confidential. Informed consent was then obtained from each participant. This study conformed to the principles of the Declaration of Helsinki. It was approved by the ethics board of the National Institute of Occupational Safety and Health of Japan (registration ID: 2021N29).

## Data analysis

In the handling situation section of the questionnaire, workers who chose to push, pull, roll, and others were excluded from the analysis; the focus was on those who chose to lift, lower, and carry, that is, to hold a load only with the body. Workers who handled loads over 55 kg were excluded from the analysis as this weight was defined as the maximum weight limit of manual handling by the International Labor Organization [24]. Moreover, workers who did not record a load weight even though they were lifting, lowering, or carrying were excluded. Continuous variables were compared between the three groups by the Kruskal-Wallis test, while dichotomous variables were compared between the three groups by the Chi-squared ($\chi^2$) test. For severe LBP, the post hoc analysis used a 2-by-2 $\chi^2$ test with Bonferroni correction.

The association between LBP (non-severe and severe) and body weight percentages (groups A, B, and C) or constant load weights (no handling, 1–5 kg, 5–10 kg, 10–15 kg, 15–20 kg, 20–25 kg, 25–30 kg, and ≥30 kg) was analyzed by multiple logistic regression using the forced entry method in which all parameters were forced into the model. Odds ratios (ORs) and 95% confidence intervals (95% CIs) were calculated for the model. The dependent variable was LBP, and the independent variable was body weight percentages or constant load weights. The adjusted variables of the model were age, body height and weight, smoking status, industry, job demand and control, worksite social support, working posture, and postural change during work. All variables were categorical data. The variance inflation factor of the independent variables had under 1.2. All statistical analyses were conducted using the Statistical Package for the Social Sciences (IBM SPSS version 27), and $p \leq 0.05$ was considered significant for all tests.

## Results

### Target of analysis

Completed questionnaires were collected from 21924 workers, including 14779 males and 7145 females (Fig 1). Of them, the male workers of groups A to C were 9607, 3623, and 1549, and the female workers were 5428, 1019, and 698, respectively.

### Basic information of workers

For male and female workers, age, body weight, and body mass index slightly differed between the three groups (Table 1). There were more workers smoking cigarettes in group C than in

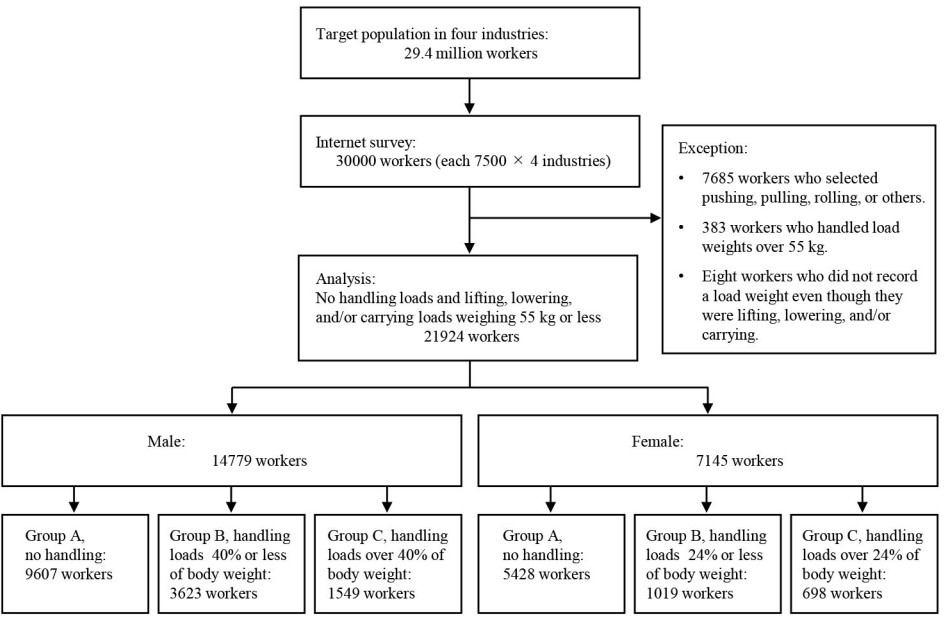

**Fig 1. Flow chart showing participation in this study.**

groups A and B. Male workers in groups A and C were many in the construction industry, while those in group B were many in the transport and postal activities. Female workers in all groups were many in the wholesale and retail trade. For male workers, full-time and day-shift accounted for approximately 90% of all groups. A high proportion of weekly working hours was 40–45 hours in groups A and B, whereas ≥50 hours in group C. For female workers, full-time accounted for approximately 50% to 60% in groups A and C, whereas approximately 40% in group B. Day-shift accounted for approximately 90%, and a high proportion of weekly working hours was <35 hours in the three groups. For both sexes, occupational stress regarding job demand was greater in group B than in group A and greater in group C than in group B. Job control was greater in groups B and C than in group A. Worksite social support for male workers was greater in group B than in groups A and C. However, in all items, the differences between the three groups were slight for male and female workers.

## Work characteristics

The number of male and female workers maintaining a neutral posture was more in group A than in group B and more in group B than in group C (Table 2). Many workers in groups B and C had restricted postural changes. During work, 35.0% of male and 24.0% of female workers were handling manual work; many male and female workers in group C performed lifting, lowering, and carrying more than in group B. The number of male and female workers handling loads for more than 2 hours per day, more than 5 times, and for a distance of more than 5 m each time was more in group C than in group B. In group B, male workers were handling loads of 1 kg or more and female workers were handling loads between 1 kg and 20 kg. In group C, male workers were handling loads of 20 kg or more and female workers were handling loads of 10 kg or more. Approximately 80% of male and female workers had no or a small effect on their work due to COVID-19.

**Table 1. Basic characteristics of workers.**

| | Male (n = 14779) | | | Female (n = 7145) | | |
|---|---|---|---|---|---|---|
| | Group A[1] (n = 9607) | Group B[2] (n = 3623) | Group C[3] (n = 1549) | Group A[1] (n = 5428) | Group B[2] (n = 1019) | Group C[3] (n = 698) |
| **(mean ± SD)** | | | | | | |
| Age (years) | $p<0.001$ | | | $p = 0.034$ | | |
| | 48.9±12.7 | 47.2±12.1 | 48.2±11.8 | 45.2±12.4 | 45.8±13.6 | 46.3±12.5 |
| Body height (cm) | $p = 0.002$ | | | $p = 0.212$ | | |
| | 170.9±6.0 | 171.0±5.8 | 170.4±5.8 | 157.9±5.6 | 157.9±5.7 | 157.5±5.8 |
| Body weight (kg) | $p<0.001$ | | | $p<0.001$ | | |
| | 69.2±11.7 | 70.1±12.3 | 65.5±9.8 | 52.8±9.2 | 54.9±10.4 | 52.3±9.8 |
| Body Mass Index | $p<0.001$ | | | $p<0.001$ | | |
| | 23.7±3.7 | 24.0±3.9 | 22.6±3.1 | 21.2±3.5 | 22.0±4.0 | 21.0±3.6 |
| **(%)** | | | | | | |
| Smoking status | $p<0.001$ | | | $p = 0.020$ | | |
| No smoking | 46.2 | 41.4 | 35.4 | 74.1 | 74.3 | 69.2 |
| Smoked in the past | 22.7 | 19.8 | 20.1 | 12.3 | 11.7 | 12.6 |
| Smoking | 31.0 | 38.8 | 44.5 | 13.6 | 14.0 | 18.2 |
| Industry | $p<0.001$ | | | $p<0.001$ | | |
| Manufacturing | 27.3 | 26.2 | 17.9 | 25.9 | 24.4 | 25.9 |
| Wholesale and retail trade | 16.8 | 20.5 | 13.4 | 37.2 | 53.8 | 42.1 |
| Construction | 32.0 | 26.1 | 36.0 | 21.7 | 8.3 | 8.0 |
| Transport and postal activities | 23.8 | 27.3 | 32.7 | 15.2 | 13.4 | 23.9 |
| Work time | $p<0.001$ | | | $p<0.001$ | | |
| Full-time | 92.4 | 88.6 | 93.3 | 63.6 | 41.0 | 52.7 |
| Part-time[4] | 7.6 | 11.4 | 6.7 | 36.4 | 59.0 | 47.3 |
| Work shift system | $p<0.001$ | | | $p<0.001$ | | |
| Day shift | 86.3 | 83.4 | 87.9 | 91.4 | 85.7 | 82.4 |
| Night, early, or late shift | 6.1 | 7.8 | 5.9 | 5.1 | 8.8 | 10.3 |
| Two or three shift | 7.6 | 8.7 | 6.1 | 3.4 | 5.5 | 7.3 |
| Total weekly working hours | $p<0.001$ | | | $p<0.001$ | | |
| <35 h | 10.5 | 11.2 | 7.9 | 35.9 | 48.9 | 39.0 |
| ≥35 h, <40 h | 21.4 | 21.1 | 17.9 | 29.6 | 26.9 | 26.5 |
| ≥40 h, <45 h | 31.7 | 28.2 | 26.6 | 24.2 | 15.9 | 19.8 |
| ≥45 h, <50 h | 16.9 | 18.0 | 18.3 | 6.2 | 4.8 | 7.3 |
| ≥50 h | 19.5 | 21.5 | 29.3 | 4.1 | 3.5 | 7.4 |
| Job demand | $p<0.001$ | | | $p<0.001$ | | |
| Low stress | 47.9 | 42.3 | 34.0 | 61.1 | 47.7 | 39.4 |
| High stress | 52.1 | 57.7 | 66.0 | 38.9 | 52.3 | 60.6 |
| Job control | $p<0.001$ | | | $p<0.001$ | | |
| Low stress | 67.2 | 59.1 | 62.9 | 62.9 | 48.8 | 50.4 |
| High stress | 32.8 | 40.9 | 37.1 | 37.1 | 51.2 | 49.6 |
| Worksite social support | $p<0.001$ | | | $p = 0.689$ | | |
| Low stress | 50.0 | 45.0 | 49.3 | 48.9 | 47.5 | 49.3 |
| High stress | 50.0 | 55.0 | 50.7 | 51.1 | 52.5 | 50.7 |

[1]No handling.

[2]Handling loads up to 40% for male workers or 24% for female workers or less of body weight.

[3]Handling loads over 40% for male workers or 24% for female workers of body weight.

[4]Part-timer and temporary worker.

**Table 2. Work characteristics.**

| | Male (n = 14779) | | | Female (n = 7145) | | |
|---|---|---|---|---|---|---|
| | Group A[1] (n = 9607) | Group B[2] (n = 3623) | Group C[3] (n = 1549) | Group A[1] (n = 5428) | Group B[2] (n = 1019) | Group C[3] (n = 698) |
| **(%)** | | | | | | |
| Working posture | $p<0.001$ | | | $p<0.001$ | | |
| Neutral posture | 65.2 | 40.4 | 29.4 | 62.8 | 43.6 | 36.4 |
| Forward-bending position | 13.2 | 16.6 | 15.5 | 18.9 | 20.9 | 21.1 |
| A half-crouching position | 6.1 | 14.9 | 18.4 | 4.8 | 12.4 | 14.8 |
| Twisting posture | 5.0 | 5.4 | 5.3 | 3.8 | 4.9 | 4.7 |
| Unstable posture | 2.9 | 4.2 | 7.4 | 1.6 | 2.9 | 4.6 |
| Other posture | 1.3 | 0.7 | 1.0 | 2.0 | 1.9 | 1.4 |
| Multiple above non-neutral postures | 6.4 | 17.8 | 23.0 | 6.0 | 13.4 | 17.0 |
| Posture change during work | $p<0.001$ | | | $p<0.001$ | | |
| Complete freedom | 49.6 | 37.5 | 37.1 | 46.5 | 37.5 | 35.4 |
| Mostly free | 29.7 | 35.9 | 35.1 | 27.5 | 29.6 | 33.7 |
| A little freedom | 14.2 | 20.2 | 20.2 | 18.6 | 21.7 | 21.9 |
| No freedom | 6.5 | 6.4 | 7.6 | 7.4 | 11.2 | 9.0 |
| Manual handling | $p<0.001$ | | | $p<0.001$ | | |
| No handling | 100 | 0 | 0 | 100 | 0 | 0 |
| Lifting and lowering | 0 | 34.9 | 24.5 | 0 | 55.4 | 43.0 |
| Carrying | 0 | 28.0 | 27.9 | 0 | 16.5 | 13.2 |
| Lifting, lowering, and carrying | 0 | 37.0 | 47.6 | 0 | 28.1 | 43.8 |
| Hours per day for handling load | $p<0.001$ | | | $p<0.001$ | | |
| <1 h | - | 23.7 | 17.0 | - | 39.1 | 29.9 |
| $\geq$1 h, <2 h | - | 36.5 | 31.6 | - | 33.6 | 31.9 |
| $\geq$2 h, <3 h | - | 13.9 | 16.1 | - | 8.5 | 11.3 |
| $\geq$3 h, <4 h | - | 9.3 | 11.4 | - | 7.3 | 10.3 |
| $\geq$4 h | - | 16.6 | 24.0 | - | 11.6 | 16.5 |
| Number of times per day to handle load by hand | $p<0.001$ | | | $p<0.001$ | | |
| <3 times | - | 38.6 | 30.2 | - | 44.1 | 36.1 |
| $\geq$3 times, <5 times | - | 13.2 | 11.7 | - | 15.2 | 13.8 |
| $\geq$5 times, <10 times | - | 16.0 | 16.5 | - | 15.1 | 17.0 |
| $\geq$10 times, <30 times | - | 20.4 | 23.3 | - | 19.5 | 21.1 |
| $\geq$30 times | - | 11.7 | 18.3 | - | 6.1 | 12.0 |
| Average carrying distance per time | $p<0.001$ | | | $p<0.001$ | | |
| No carrying | - | 22.2 | 14.7 | - | 40.8 | 30.9 |
| $\geq$1 m, < 5m | - | 41.8 | 39.3 | - | 42.2 | 45.4 |
| $\geq$5 m, <10 m | - | 19.5 | 24.5 | - | 9.0 | 14.6 |
| $\geq$10 m, <20 m | - | 8.5 | 11.1 | - | 3.8 | 4.2 |
| $\geq$20 m | - | 8.0 | 10.5 | - | 4.1 | 4.9 |
| Constant load weights | $p<0.001$ | | | $p<0.001$ | | |
| No handling | 100 | 0 | 0 | 100 | 0 | 0 |
| $\geq$1 kg, <5 kg | 0 | 4.0 | 0 | 0 | 14.1 | 0 |
| $\geq$5 kg, <10 kg | 0 | 12.5 | 0 | 0 | 36.9 | 0 |
| $\geq$10 kg, <15 kg | 0 | 24.4 | 0 | 0 | 45.4 | 7.7 |
| $\geq$15 kg, <20 kg | 0 | 12.2 | 0 | 0 | 3.2 | 24.4 |
| $\geq$20 kg, <25 kg | 0 | 31.5 | 1.2 | 0 | 0 | 35.4 |
| $\geq$25 kg, <30 kg | 0 | 8.1 | 9.4 | 0 | 0 | 9.3 |

*(Continued)*

**Table 2.** (Continued)

| | Male (n = 14779) | | | Female (n = 7145) | | |
|---|---|---|---|---|---|---|
| | Group A[1] (n = 9607) | Group B[2] (n = 3623) | Group C[3] (n = 1549) | Group A[1] (n = 5428) | Group B[2] (n = 1019) | Group C[3] (n = 698) |
| ≥30 kg | 0 | 7.3 | 89.5 | 0 | 0 | 23.2 |
| Impact of COVID-19 | $p<0.001$ | | | $p<0.001$ | | |
| No impact | 45.0 | 41.7 | 39.4 | 51.3 | 48.9 | 44.3 |
| A little less work | 30.5 | 32.3 | 32.5 | 28.0 | 25.2 | 25.4 |
| A little more work | 8.3 | 7.0 | 7.4 | 7.6 | 10.1 | 12.3 |
| Much less work | 13.6 | 16.3 | 16.7 | 10.9 | 12.1 | 14.2 |
| Much more work | 2.6 | 2.6 | 4.1 | 2.2 | 3.7 | 3.9 |

[1]No handling.

[2]Handling loads up to 40% for male workers or 24% for female workers or less of body weight.

[3]Handling loads over 40% for male workers or 24% for female workers of body weight.

### Prevalence of severe LBP

In groups A, B, and C, 25.5%, 39.2%, and 47.3% of male workers or 16.9%, 26.4%, and 38.0% of female workers had severe LBP, respectively. The prevalence of severe LBP among male and female workers was greater in group B than in group A ($p < 0.001$) and greater in group C than in group B ($p < 0.001$). Of severe LBP, 20.1% of male workers had occurred less than 1 year ago, 27.4% did 1–5 years ago, and 52.5% did more than 5 years ago; 25.0% of female workers had occurred less than 1 year ago, 32.7% did 1–5 years ago, and 42.2% did more than 5 years ago.

### Association between severe LBP with weight limit as body weight percentages

Table 3 summarizes the association between severe LBP and weight limit as body weight percentages, as revealed by the multiple logistic regression models. The ORs of severe LBP for male and female workers were significantly greater in group B (male's OR: 1.40 and female's OR:1.26) than in group A and even greater in group C (1.74 and 2.06).

### Association between severe LBP with weight limit as constant load weights

Table 4 summarizes the association between severe LBP and weight limit as constant load weights, as revealed by the multiple logistic regression models. The ORs of severe LBP among male and female workers handling loads of 10 kg or more were significantly greater than those of no weight handling (male's OR: 1.25 to 1.75 and female's OR: 1.46 to 3.00) and increased with each additional weight category. However, there was no significant difference between no handling and handling loads of under 10 kg in male and female workers.

## Discussion

This study aimed to clarify the effect of relative weight limits set as body weight percentages on LBP prevalence. For body weight percentages, the prevalence of severe LBP was greater in group B than in group A but lesser than in group C. These associations were similar for the adjusted ORs of LBP. For constant load weights, the ORs of LBP among male and female workers handling loads under 10 kg were not significantly different from no handling.

**Table 3. Association of severe low back pain with weight limit as body weight percentages from multiple logistic regression.**

| | Male (n = 14779) | | Female (n = 7145) | |
|---|---|---|---|---|
| | OR (95%CI) | p-value | OR (95%CI) | p-value |
| Body weight percentages | | | | |
| Group A[1] | 1.00 (Reference) | | 1.00 (Reference) | |
| Group B[2] | 1.40 (1.28–1.53) | <0.001 | 1.26 (1.06–1.49) | 0.008 |
| Group C[3] | 1.74 (1.54–1.97) | <0.001 | 2.06 (1.71–2.48) | <0.001 |
| **(Adjusted variables)** | | | | |
| Age | | | | |
| 20–39 y | 1.00 (Reference) | | 1.00 (Reference) | |
| 40–49 y | 1.78 (1.59–1.99) | <0.001 | 1.42 (1.21–1.68) | <0.001 |
| 50–59 y | 2.25 (2.01–2.51) | <0.001 | 1.58 (1.33–1.88) | <0.001 |
| 60–75 y | 2.65 (2.35–2.99) | <0.001 | 1.75 (1.45–2.12) | <0.001 |
| Body height | | | | |
| <160 cm | 1.00 (Reference) | | 1.00 (Reference) | |
| ≥160 cm, <170 cm | 0.89 (0.67–1.19) | 0.437 | 1.29 (1.13–1.48) | <0.001 |
| ≥170 cm, <180 cm | 0.97 (0.73–1.30) | 0.846 | 1.69 (1.15–2.48) | 0.008 |
| ≥180 cm | 1.10 (0.80–1.50) | 0.564 | 0.84 (0.15–4.77) | 0.840 |
| Body weight | | | | |
| <50 kg | 1.00 (Reference) | | 1.00 (Reference) | |
| ≥50 kg, <60 kg | 1.52 (1.02–2.27) | 0.038 | 1.18 (1.02–1.37) | 0.023 |
| ≥60 kg, <70 kg | 1.77 (1.19–2.62) | 0.005 | 1.30 (1.08–1.58) | 0.007 |
| ≥70 kg, <80 kg | 2.03 (1.36–3.02) | <0.001 | 1.73 (1.28–2.34) | <0.001 |
| ≥80 kg | 2.06 (1.38–3.08) | <0.001 | 1.76 (1.17–2.65) | 0.007 |
| Smoking status | | | | |
| No smoking | 1.00 (Reference) | | 1.00 (Reference) | |
| Smoked in the past | 1.09 (0.98–1.20) | 0.114 | 1.16 (0.96–1.39) | 0.123 |
| Smoking | 1.41 (1.29–1.54) | <0.001 | 1.34 (1.13–1.59) | <0.001 |
| Industry | | | | |
| Manufacturing | 1.00 (Reference) | | 1.00 (Reference) | |
| Wholesale and retail trade | 1.00 (0.89–1.13) | 0.980 | 1.10 (0.94–1.29) | 0.237 |
| Construction | 0.97 (0.87–1.07) | 0.510 | 1.11 (0.91–1.36) | 0.287 |
| Transport and postal activities | 1.01 (0.91–1.13) | 0.811 | 1.10 (0.90–1.34) | 0.363 |
| Job demand | | | | |
| Low stress | 1.00 (Reference) | | 1.00 (Reference) | |
| High stress | 1.47 (1.35–1.59) | <0.001 | 1.50 (1.32–1.70) | <0.001 |
| Job control | | | | |
| Low stress | 1.00 (Reference) | | 1.00 (Reference) | |
| High stress | 1.05 (0.97–1.14) | 0.241 | 1.11 (0.98–1.27) | 0.107 |
| Worksite social support | | | | |
| Low stress | 1.00 (Reference) | | 1.00 (Reference) | |
| High stress | 0.99 (0.91–1.07) | 0.745 | 1.05 (0.93–1.20) | 0.437 |
| Working posture | | | | |
| Neutral posture | 1.00 (Reference) | | 1.00 (Reference) | |
| Forward-bending position | 1.74 (1.56–1.94) | <0.001 | 1.80 (1.54–2.12) | <0.001 |
| A half-crouching position | 2.87 (2.53–3.26) | <0.001 | 3.03 (2.44–3.76) | <0.001 |
| Twisting posture | 2.13 (1.80–2.51) | <0.001 | 2.34 (1.77–3.09) | <0.001 |
| Unstable posture | 2.00 (1.65–2.42) | <0.001 | 2.56 (1.78–3.70) | <0.001 |
| Other posture | 2.57 (1.86–3.56) | <0.001 | 2.39 (1.63–3.49) | <0.001 |

*(Continued)*

**Table 3.** (Continued)

| | Male (n = 14779) | | Female (n = 7145) | |
|---|---|---|---|---|
| | OR (95%CI) | *p*-value | OR (95%CI) | *p*-value |
| Multiple above non-neutral postures | 3.28 (2.90–3.70) | <0.001 | 3.41 (2.79–4.18) | <0.001 |
| Posture change during work | | | | |
| Completely freedom | 1.00 (Reference) | | 1.00 (Reference) | |
| Mostly freedom | 1.41 (1.29–1.54) | <0.001 | 1.45 (1.25–1.70) | <0.001 |
| A little freedom | 2.23 (1.99–2.49) | <0.001 | 2.11 (1.79–2.50) | <0.001 |
| No freedom | 2.30 (1.98–2.68) | <0.001 | 1.95 (1.56–2.45) | <0.001 |

[1]No handling.

[2]Handling loads up to 40% for male workers or 24% for female workers or less of body weight.

[3]Handling loads over 40% for male workers or 24% for female workers of body weight.

Limiting weight to 40% for male workers and 24% for female workers or less of body weight had some preventive effects on LBP. However, the preventive effects on LBP caused by handling loads were insufficient since their LBP risk was greater than those of no handling loads. From a biomechanical point of view, limiting weights as body weight percentages are problematic. When people of the same height and different weights take the same posture, the compression and shearing forces on the lumbar disc are greater in a heavier than in a lighter person [18, 19]. This is because the extra weight in a heavier person increases the compression and shearing forces on the lumbar disc. In addition, with the weight limit set as body weight percentages, a heavier person can lift heavier loads, increasing the compression and shear forces on the lumbar disc. Conversely, limiting weight under 10 kg suppressed the development of LBP caused by handling loads. Therefore, relative weight limits set as body weight percentages were inappropriate and ineffective for preventing LBP. Workers' LBP may be suppressed by an absolute load weight rather than a relative load weight.

In this study, a load weight of 10 kg was the threshold for LBP caused by handling loads. Palmer et al. (2003) [25] reported that male and female workers lifting loads of 10 kg or more had greater prevalence ratios of LBP and sciatica than those under 10 kg. Hoogendoorn et al. (2002) [26] revealed that workers handling loads of 10 kg or more took more sickness leaves for three days or longer due to LBP than those who did not handle any weight. Macfarlane et al. (1997) [27] found that the OR of LBP among female workers lifting or moving loads over 25 lb (11.3 kg) was greater than those handling loads of 25 lb or lesser. Nahit et al. (2001) [28] reported that the OR of LBP among workers lifting loads of 25 lb (11.3 kg) with one hand and 50 lb (22.7 kg) with two hands was greater than those of the other workers. These findings are similar to the results of this study. Conversely, several studies found no association between lifting or carrying loads and LBP prevalence [29–31]. Prado-Leon et al. (2005) [32] reported that driving and lifting were associated with LBP, but only lifting was not associated with it. The previous studies differ in the targeted occupational groups, work content, and evaluation criterion of LBP; hence, positive and negative results may have been obtained. Although further studies are needed to clarify the association between the value of load weight and LBP, under 10 kg obtained in this study may be a reasonable result even for Japanese workers to prevent LBP.

In ISO 11228–1, the weight limit for males is 25 kg and for females is 20 kg. The recommended mass limit (RML) is calculated according to work content such as horizontal and vertical location, travel distance, twisting angle, frequency and duration of lifting, and grasp type based on the weight limits [15]. RML is smaller than 25 kg or 20 kg because it is calculated by

**Table 4. Association of severe low back pain with weight limit as constant load weights from multiple logistic regression.**

| | Male (n = 14779) | | Female (n = 7145) | |
|---|---|---|---|---|
| | OR (95%CI) | *p*-value | OR (95%CI) | *p*-value |
| Constant load weights | | | | |
| No handling | 1.00 (Reference) | | 1.00 (Reference) | |
| ≥1 kg, <5 kg | 0.95 (0.63–1.43) | 0.788 | 1.33 (0.89–2.01) | 0.166 |
| ≥5 kg, <10 kg | 0.94 (0.75–1.18) | 0.587 | 1.05 (0.81–1.38) | 0.704 |
| ≥10 kg, <15 kg | 1.25 (1.07–1.46) | 0.005 | 1.46 (1.18–1.82) | <0.001 |
| ≥15 kg, <20 kg | 1.54 (1.25–1.90) | <0.001 | 1.51 (1.09–2.08) | 0.012 |
| ≥20 kg, <25 kg | 1.62 (1.41–1.85) | <0.001 | 1.89 (1.41–2.52) | <0.001 |
| ≥25 kg, <30 kg | 1.58 (1.28–1.95) | <0.001 | 3.00 (1.77–5.11) | <0.001 |
| ≥30 kg | 1.75 (1.56–1.97) | <0.001 | 2.67 (1.89–3.76) | <0.001 |
| **(Adjusted variables)** | | | | |
| Age | | | | |
| 20–39 y | 1.00 (Reference) | | 1.00 (Reference) | |
| 40–49 y | 1.75 (1.57–1.96) | <0.001 | 1.42 (1.21–1.68) | <0.001 |
| 50–59 y | 2.21 (1.97–2.47) | <0.001 | 1.58 (1.33–1.88) | <0.001 |
| 60–75 y | 2.61 (2.32–2.95) | <0.001 | 1.75 (1.45–2.12) | <0.001 |
| Body height | | | | |
| <160 cm | 1.00 (Reference) | | 1.00 (Reference) | |
| ≥160 cm, <170 cm | 0.90 (0.68–1.20) | 0.479 | 1.30 (1.14–1.48) | <0.001 |
| ≥170 cm, <180 cm | 0.98 (0.74–1.32) | 0.915 | 1.71 (1.16–2.52) | 0.007 |
| ≥180 cm | 1.11 (0.81–1.53) | 0.505 | 0.80 (0.14–4.67) | 0.804 |
| Body weight | | | | |
| <50 kg | 1.00 (Reference) | | 1.00 (Reference) | |
| ≥50 kg, <60 kg | 1.50 (1.01–2.23) | 0.047 | 1.18 (1.02–1.36) | 0.029 |
| ≥60 kg, <70 kg | 1.71 (1.16–2.54) | 0.007 | 1.27 (1.05–1.54) | 0.014 |
| ≥70 kg, <80 kg | 1.95 (1.31–2.90) | <0.001 | 1.70 (1.25–2.29) | <0.001 |
| ≥80 kg | 1.96 (1.31–2.93) | 0.001 | 1.68 (1.12–2.53) | 0.013 |
| Smoking status | | | | |
| No smoking | 1.00 (Reference) | | 1.00 (Reference) | |
| Smoked in the past | 1.09 (0.98–1.20) | 0.108 | 1.16 (0.97–1.40) | 0.109 |
| Smoking | 1.41 (1.29–1.53) | <0.001 | 1.34 (1.13–1.59) | <0.001 |
| Industry | | | | |
| Manufacturing | 1.00 (Reference) | | 1.00 (Reference) | |
| Wholesale and retail trade | 1.01 (0.89–1.14) | 0.906 | 1.09 (0.94–1.28) | 0.261 |
| Construction | 0.96 (0.87–1.06) | 0.433 | 1.10 (0.90–1.34) | 0.342 |
| Transport and postal activities | 1.01 (0.91–1.12) | 0.863 | 1.07 (0.88–1.30) | 0.519 |
| Job demand | | | | |
| Low stress | 1.00 (Reference) | | 1.00 (Reference) | |
| High stress | 1.46 (1.35–1.58) | <0.001 | 1.49 (1.32–1.69) | <0.001 |
| Job control | | | | |
| Low stress | 1.00 (Reference) | | 1.00 (Reference) | |
| High stress | 1.06 (0.97–1.15) | 0.205 | 1.11 (0.97–1.26) | 0.119 |
| Worksite social support | | | | |
| Low stress | 1.00 (Reference) | | 1.00 (Reference) | |
| High stress | 0.99 (0.92–1.07) | 0.808 | 1.05 (0.93–1.19) | 0.458 |
| Working posture | | | | |
| Neutral posture | 1.00 (Reference) | | 1.00 (Reference) | |

(*Continued*)

**Table 4.** (Continued)

| | Male (n = 14779) | | Female (n = 7145) | |
|---|---|---|---|---|
| | OR (95%CI) | *p*-value | OR (95%CI) | *p*-value |
| Forward-bending position | 1.73 (1.55–1.93) | <0.001 | 1.81 (1.54–2.13) | <0.001 |
| A half-crouching position | 2.84 (2.50–3.23) | <0.001 | 3.00 (2.42–3.73) | <0.001 |
| Twisting posture | 2.11 (1.79–2.49) | <0.001 | 2.37 (1.79–3.13) | <0.001 |
| Unstable posture | 2.00 (1.65–2.41) | <0.001 | 2.50 (1.72–3.61) | <0.001 |
| Other posture | 2.57 (1.86–3.55) | <0.001 | 2.40 (1.64–3.52) | <0.001 |
| Multiple above non-neutral postures | 3.24 (2.86–3.66) | <0.001 | 3.43 (2.80–4.19) | <0.001 |
| Posture change during work | | | | |
| Completely freedom | 1.00 (Reference) | | 1.00 (Reference) | |
| Mostly freedom | 1.41 (1.29–1.54) | <0.001 | 1.46 (1.25–1.71) | <0.001 |
| A little freedom | 2.23 (2.00–2.49) | <0.001 | 2.14 (1.81–2.53) | <0.001 |
| No freedom | 2.32 (2.00–2.71) | <0.001 | 1.96 (1.56–2.46) | <0.001 |

multiplying the maximum load by nine multipliers from 0 to 1. When operating with a high workload, the RML calculated is lower. In this study, the suggested 10 kg corresponds to RML, not weight limits, because the weight value was derived from work involving various operations. A weight limit for Japanese workers should be considered in the future.

This study had several limitations. First, the web-based questionnaire was answered by workers registered with monitor research companies; hence, sampling bias may have affected the results. Respondents of 30000 were obtained, corresponding to approximately 0.1% of the target population, to lower the bias. Second, the survey was conducted while COVID-19 had not subsided; therefore, this result may include the impact of labor shortages or reduced orders due to COVID-19. However, only a few workers had a considerable effect of COVID-19 on their working style. Finally, as results were described by recalling past work and physical conditions, they may have been affected by recall bias. In addition, this study is a cross-sectional survey; therefore, causal associations cannot be determined. Further studies must be conducted that would consider these points.

## Conclusion

In conclusion, setting relative weight limits as body weight percentages could not adequately prevent LBP. However, handling loads under 10 kg could suppress the prevalence of LBP. Therefore, guidelines to prevent LBP among workers should not be limited to relative body weight percentages.

## Supporting information

**S1 Dataset.**
(XLSX)

## Acknowledgments

The authors would like to thank the participants for their kind contributions.

## Author Contributions

**Conceptualization:** Kazuyuki Iwakiri, Takeshi Sasaki, Midori Sotoyama, Tanghuizi Du, Keiichi Miki.

**Data curation:** Kazuyuki Iwakiri, Takeshi Sasaki.

**Formal analysis:** Kazuyuki Iwakiri, Takeshi Sasaki.

**Funding acquisition:** Kazuyuki Iwakiri.

**Investigation:** Kazuyuki Iwakiri, Takeshi Sasaki.

**Methodology:** Kazuyuki Iwakiri, Takeshi Sasaki, Midori Sotoyama, Tanghuizi Du.

**Supervision:** Midori Sotoyama.

**Validation:** Kazuyuki Iwakiri.

**Writing – original draft:** Kazuyuki Iwakiri.

**Writing – review & editing:** Kazuyuki Iwakiri, Takeshi Sasaki, Midori Sotoyama, Tanghuizi Du, Keiichi Miki.

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
