## [Decision Letter · Decision Letter 0]

3 Feb 2023

PONE-D-22-34000Effect of weight limit set as a body weight percentage on work-related low back pain among workersPLOS ONE

Dear Dr. Iwakiri,

Thank you for submitting your manuscript to PLOS ONE. After careful consideration, we feel that it has merit but does not fully meet PLOS ONE’s publication criteria as it currently stands. Therefore, we invite you to submit a revised version of the manuscript that addresses the points raised during the review process.

We look forward to receiving your revised manuscript.

Kind regards,

Yee Guan Ng, Ph.D

Academic Editor

PLOS ONE

Journal Requirements:

Additional Editor Comments:

The results of this study including the methodology employed is a very interesting study.

However, I particularly concur with comments from reviewer one.

Please carry out the necessary improvement as per comments provided.

In addition, I am also intrigue with the methodology employed.

Specifically, why was the load weight classified?

What was the reference for the brackets of weight category?

In addition, how did the analysis take into consideration of the severity of LBP?

Reviewers' comments:

Reviewer's Responses to Questions

**Comments to the Author**

1. Is the manuscript technically sound, and do the data support the conclusions?

Reviewer #1: Partly

Reviewer #2: Yes

2. Has the statistical analysis been performed appropriately and rigorously? 

Reviewer #1: Yes

Reviewer #2: Yes

3. Have the authors made all data underlying the findings in their manuscript fully available?

Reviewer #1: No

Reviewer #2: Yes

4. Is the manuscript presented in an intelligible fashion and written in standard English?

Reviewer #1: No

Reviewer #2: Yes

5. Review Comments to the Author

Reviewer #1: Many valuable information was collected with an aim to understand the association between LBP and lifting risk variables. Although the authors examined the effectiveness of the current Japanese lifting standard using a percentage of body weight as the weight limit in reducing LBP, the authors did not explore other risk estimates using ISO or NIOSH methods. Most discussion does not add any value to the current science. What the study could have done is to use various risk calculation methods to determine which combination of the risk variables in the dataset is associated with the prevalence of LBP. The author could have even used some of the posture variables in combination of the weight limit as evidence for supporting the association. The suggested new findings may be used as the base for revising the Japan’s inadequate lifting standard using a percentage of body weight as the limit. The manuscript does have some merit that is worth publishing after a major revision that addresses the above general comment and the below specific comments.

Line 24: the sentence “Moreover,……” in unclear. What are the categories of the constant load weights?

Line 111: Ten items are listed for examining job demands. If ten items were used, the lowest possible scores for job demands should be 10 assuming the Likert scale 1-4 was used. No variables are listed for job control and social support. Please add the info because these two additional variables were used in the study. What was the cut-off point of the score used for low and high stress? Please clarify.

Line 115: For clarity, I suggest the sentence from lines 92-93 for LBP be moved to LBP section (Lines 114-122).

Line 124-128: Please specify if the posture and postural changes are for the trunk or torso, otherwise, the postures asked could mean hand, arm, and other body joints.

Lines 155-158: Excluding the participants that performed push/pull or lifting more than 55 kgs is not justified well. Although there are recommended weight limits in various standard, participants who lifted more than 55 kg should be included in the analysis. Excluding pushing/pulling tasks is less concerning.

Line 167: Please explain what “forced entry metho” is? Add the reference groups for all the variables to the text, unless Tables have the info. Add a description for the dependent variables, such as linear or categorical multiple logistic regression. Technically the constant load weight could be modeled as linear continuous variables for their linear intervals.

Line 178-183: Text may be removed because all the presented info is in Figure 1. Suggest that the authors simple describe the total sample size that met the inclusion criteria.

Table 1: Change “body height” to statue. Ad a footnote to define part-time. Recommend that the authors add % or mean to each variable because they are not very obvious. The scores of the job demands, control and social support have little clinical meaning. Suggest that the author change them to low and high using the medium cut-off point or Japanese stress standard in the literature for classifying the stress level. Then percentage of low or high level for each variable can be presented. Chi-square test can be further used for comparing pairs of the different levels of each variables. For example, the job control in female Group A may be different from Group B and C, but there is no difference between Groups B and C.

Line 209: Improper posture is not an academic term. Suggest that the authors use non-neutral posture if this is what it meant in the questionnaire. A larger number sounds better than a higher number. Suggest that the authors change the expression throughout the manuscript. Chang the section title to “Work characteristics”.

Line 228: Because the first episode of the LBP was recorded in a retrospective manner regardless of the length of recall time (i.e., could be one month or 10 years). It may be inappropriate to describe the reporting of LBP as simply prevalence. Consider using a revised term career-time cumulative prevalence of LBP.

Table 3 title is misleading because only the bottom three lines have the body weight percentage limit information. The reference group for each variable is different and nothing to do with the body weight limit. Table 3 needs to be revised to show proper comparisons. Similarly, Table 4 needs to be revised according to the above question.

Line 266: The statement “For percentages of body weight, the prevalence of severe LBP was lower in group B than in group C but higher than in group A. These associations were similar for the adjusted ORs of LBP.” May be the most important finding of the study. However, no discussion is provided for why the moderate load in term of body weight limit had the highest prevalence of LBP. The second paragraph in Discussion provides little interpretation of the results. It reads like results. The third paragraph discusses BMI, but the preceding paragraph (2nd) does not have BMI info. BMI data do support the argument. The higher BMI in Group B should be used for discussion.

Line 288: The sentence “This study showed that handling loads of 10 kg or more could not suppress severe LBP.” does not make sense. The following discussion shows mixed effect of lifted weight on the development of LBP. Both ISO and NIOSH use lifted load and postural information to determine the risk of LBP. It’s a scientific consensus that both are important lifting risk information. Using load alone is not a good risk estimate.

Reviewer #2: This study examined whether weight-based transport restrictions, which have been considered useful for the prevention of back pain in working people, are effective.

We consider your research method and the conclusions drawn to be valid and very significant.

The following comments should be reviewed and answered.

１）Abstract:

Gender representation is mixed in terms of Men/women and Male/Female. It would be better to unify them.

２）Basic information of workers

Duration of occupational engagement may influence the onset of low back pain. The authors should add to Basic information if possible.

３）Work description

　Line 220-221：「In the three groups, 82.7% of male and 86.0% of female workers had no or a small effect due to COVID-19 on their work」Data showing this statement is not included in the table 2, it would be better to add it.

6. PLOS authors have the option to publish the peer review history of their article (what does this mean?). If published, this will include your full peer review and any attached files.

Reviewer #1: No

Reviewer #2: No

---

## [Author Response · Author response to Decision Letter 0]

15 Mar 2023

The "Response to reviewers" file lists all comments' responses.

---

## [Decision Letter · Decision Letter 1]

3 Apr 2023

Effect of relative weight limit set as a body weight percentage on work-related low back pain among workers

PONE-D-22-34000R1

Dear Dr. Iwakiri,

We’re pleased to inform you that your manuscript has been judged scientifically suitable for publication and will be formally accepted for publication once it meets all outstanding technical requirements.

Kind regards,

Yee Guan Ng, Ph.D

Academic Editor

PLOS ONE

Additional Editor Comments (optional):

Dear Authors,

Thank you very much for your patience.

After a thorough review, your revised paper in addressing all the comments made by reviewers is recommended for publication to the Editorial Board.

Reviewers' comments:

Reviewer's Responses to Questions

**Comments to the Author**

1. If the authors have adequately addressed your comments raised in a previous round of review and you feel that this manuscript is now acceptable for publication, you may indicate that here to bypass the “Comments to the Author” section, enter your conflict of interest statement in the “Confidential to Editor” section, and submit your "Accept" recommendation.

Reviewer #1: All comments have been addressed

Reviewer #2: All comments have been addressed

2. Is the manuscript technically sound, and do the data support the conclusions?

Reviewer #1: Yes

Reviewer #2: Yes

3. Has the statistical analysis been performed appropriately and rigorously? 

Reviewer #1: Yes

Reviewer #2: Yes

4. Have the authors made all data underlying the findings in their manuscript fully available?

Reviewer #1: Yes

Reviewer #2: Yes

5. Is the manuscript presented in an intelligible fashion and written in standard English?

Reviewer #1: Yes

Reviewer #2: Yes

6. Review Comments to the Author

Reviewer #1: All comments are addressed appropriately. The statistical analysis was performed appropriately. The manuscript is presented clearly.

Reviewer #2: The authors have made appropriate corrections. No further amendments are required from me. This paper is well worth accepting.

7. PLOS authors have the option to publish the peer review history of their article (what does this mean?). If published, this will include your full peer review and any attached files.

Reviewer #1: No

Reviewer #2: No

---

## [Editor Report · Acceptance letter]

11 Apr 2023

PONE-D-22-34000R1 

Effect of relative weight limit set as a body weight percentage on work-related low back pain among workers 

Dear Dr. Iwakiri:

I'm pleased to inform you that your manuscript has been deemed suitable for publication in PLOS ONE. Congratulations! Your manuscript is now with our production department. 

Kind regards, 

on behalf of

Dr. Yee Guan Ng 

Academic Editor

PLOS ONE